# Lamotrigine Attenuates Neuronal Excitability, Depresses GABA Synaptic Inhibition, and Modulates Theta Rhythms in Rat Hippocampus

**DOI:** 10.3390/ijms222413604

**Published:** 2021-12-19

**Authors:** Paulina Kazmierska-Grebowska, Marcin Siwiec, Joanna Ewa Sowa, Bartosz Caban, Tomasz Kowalczyk, Renata Bocian, M. Bruce MacIver

**Affiliations:** 1Department of Neurobiology, Faculty of Biology and Environmental Protection, University of Lodz, 90-236 Lodz, Poland; bartosz.caban@biol.uni.lodz.pl (B.C.); tomasz.kowalczyk1@biol.uni.lodz.pl (T.K.); renata.bocian@biol.uni.lodz.pl (R.B.); 2Department of Physiology, Institute of Pharmacology, Polish Academy of Sciences, 31-343 Krakow, Poland; siwiec@if-pan.krakow.pl (M.S.); joasowa@if-pan.krakow.pl (J.E.S.); 3Department of Anesthesia, Stanford University School of Medicine, Stanford, CA 94305, USA

**Keywords:** I*h* current, lamotrigine, HCN channels, theta rhythm, local field potentials, IPSCs, membrane excitability, action potential, membrane resonance

## Abstract

Theta oscillations generated in hippocampal (HPC) and cortical neuronal networks are involved in various aspects of brain function, including sensorimotor integration, movement planning, memory formation and attention. Disruptions of theta rhythms are present in individuals with brain disorders, including epilepsy and Alzheimer’s disease. Theta rhythm generation involves a specific interplay between cellular (ion channel) and network (synaptic) mechanisms. HCN channels are theta modulators, and several medications are known to enhance their activity. We investigated how different doses of lamotrigine (LTG), an HCN channel modulator, and antiepileptic and neuroprotective agent, would affect HPC theta rhythms in acute HPC slices (in vitro) and anaesthetized rats (in vivo). Whole-cell patch clamp recordings revealed that LTG decreased GABA_A_-fast transmission in CA3 cells, in vitro. In addition, LTG directly depressed CA3 and CA1 pyramidal neuron excitability. These effects were partially blocked by ZD 7288, a selective HCN blocker, and are consistent with decreased excitability associated with antiepileptic actions. Lamotrigine depressed HPC theta oscillations in vitro, also consistent with its neuronal depressant effects. In contrast, it exerted an opposite, enhancing effect, on theta recorded in vivo. The contradictory in vivo and in vitro results indicate that LTG increases ascending theta activating medial septum/entorhinal synaptic inputs that over-power the depressant effects seen in HPC neurons. These results provide new insights into LTG actions and indicate an opportunity to develop more precise therapeutics for the treatment of dementias, memory disorders and epilepsy.

## 1. Introduction

The theta rhythm in mammalian limbic cortex is a prime example of a bioelectric oscillatory pattern requiring central mechanisms for oscillation and synchronization [1,2,3,4]. Theta waves are recorded as sinusoidal, high-voltage activity (from 0.2 mV to 2 mV) with a frequency band ranging from 3 Hz to 12 Hz in rodents [1,2,4] and have been extensively used to study neural network excitability [3]. Theta rhythms occur in mammalian hippocampus during the planning and initiation of movement sequences [5], memory formation [6], and are involved in attention and sensorimotor integration [2,7]. Theta power has been shown to increase before the onset of convulsive seizures in kainic acid-induced animal models of epilepsy [8].

The appearance of theta rhythms in HPC neuronal networks closely correlates with theta-ON and theta-OFF cell discharge in CA1 and DG subfields [9]. Intracellular biophysical mechanisms reflected in rhythmic membrane potential oscillations (MPOs) are necessary for the appearance of synchronous field activity in the theta band [9,10,11]. Voltage-gated sodium and calcium channels [1], as well as HCN channels, contribute to MPO generation [11,12,13,14]. Some studies note the participation of IPSPs [15,16,17], and others suggest the involvement of EPSPs in theta generation [18,19].

HCN channels are members of a family of voltage-gated ion channels that mediate a hyperpolarization activated current (I*h*), an inward cationic current activated upon hyperpolarization of the membrane potential. They contribute to intrinsic excitability, pacemaker activity, and synaptic integration [12,14,20,21,22]. HCN channels are tetrameric structures consisting of either homomeric or heteromeric complexes of pore-forming subunits (HCN1-HCN4) [22]. The expression patterns of the four HCN subunits appear to overlap to assemble heteromeric complexes in some neurons, including HPC pyramidal neurons [23]. HCN channels, in particular HCN1 subunits, are a central modulator of HPC -based memory [24,25]. Microiontophoretic blockage of HCN channels reduces the discharge frequency and perturbs theta rhythms [26]. Bender et al. 2003 [27] found a pivotal role for HCN channels expressed on CA3 pyramidal neurons in giant depolarizing potential-related network synchronization; however, no direct studies have verified the role of I*h* in theta rhythmogenesis.

HCN channels are nonspecifically opened by lamotrigine (LTG), a commonly prescribed anti-epileptic drug [28,29]. In addition to its effects on HCN channels, LTG blocks voltage-gated Na^+^ channels, stabilizes their inactive state and blocks Ca^2+^ channels [30], as well as enhances I*h* mediated by HCN channels in pyramidal neurons [31]. LTG has been used in therapy for amyotrophic lateral sclerosis and Parkinson’s disease [32]. It was also shown to improve word recognition, naming, and mood in patients with symptoms of Alzheimer’s disease [33]. LTG was shown to significantly attenuate pyramidal neuron damage resulting from ischemia [34] and to rescue electrophysiological and cognitive deficits in two independent neurofibromatosis type 1 mouse models [35].

Since HCN channels are involved in the regulation of neuronal membrane properties and pacemaker activity, we aimed to directly verify the role of I*h* in the modulation of HPC theta oscillations in rodents. We set out to assess the direct effect of LTG administration on pyramidal cell membrane properties using acute in vitro whole-cell patch-clamp recordings to elucidate the effects of LTG on the HPC neurons. It remains controversial whether LTG increases [36] or decreases [37] synaptic inhibition, so we carefully examined this as well. We then studied LTG effects on theta rhythms in vitro and in vivo, in rats. Our working hypothesis was that the membrane and synaptic effects produced by LTG would manifest themselves as alterations of theta rhythms.

## 2. Results

### 2.1. Effect of LTG on Spontaneous GABAergic Synaptic Activity in CA3c Pyramidal Cells

We were particularly interested in determining whether LTG affects GABAergic synaptic transmission in the HPC al CA3c region. At a holding potential of 0 mV, spontaneous inhibitory postsynaptic (sIPSCs) were recorded as outward currents. Lamotrigine (50 μM) reduced the median sIPSC frequency (from 7.05 ±1.48 to 4.05 ± 0.86 Hz, number of slices = 5, *t* = 4.34, df = 4, *p* = 0.01, paired *t*-test; Figure 1). Moreover, the amplitude of sIPSCs was also decreased (baseline: 35.24 ± 4.03 pA; LTG: 25.26 2.29 pA; number of slices = 5, *t* = 5.22, df = 4, *p* = 0.01, paired *t*-test) without altering sIPSC kinetics (data not shown). ZD 7288 at a concentration known to efficiently block HCN channels (10 μM) [38,39,40,41,42], reduced the relative LTG-induced change in sIPSC frequency: 102.2 ± 4.65%, V = 8, *p* = 1, paired Wilcoxon signed rank test; amplitude: 95.84 ± 4.03, number of slices = 5, V = 11, *p* = 0.44, paired Wilcoxon signed rank test, suggesting that LTG’s effects depend on HCN activation.

sIPSCs are a mixture of both miniature events (mIPSCs) and currents evoked by spontaneous action potentials in presynaptic cells. TTX (500 nM) was used to block neuronal firing, and the mIPSC frequency, amplitude, and kinetics were measured before and after LTG application to determine whether the decrease in sIPSC frequency induced by LTG was due to a change in the spontaneous action potential firing of interneurons. The frequency of mIPSCs was 1.8 ± 0.35 Hz, much lower than sIPSCs (7.05 ± 1.48 Hz, *t* = 3.45, df = 4.44, *p* = 0.02, *t*-test), indicating that most sIPSCs were driven by action potentials. Neither the frequency nor amplitude of the mIPSC were affected by LTG application (mIPSC frequency: 136.7 ± 28.1%, *t* = −1.58, df = 6, *p* = 0.17, paired *t*-test; amplitude: 102.8 ± 4.09, number of slices = 7, *t* = −0.71, df = 6, *p* = 0.5, paired *t*-test), indicating that LTG depressed the excitability of interneurons.

### 2.2. Effect of LTG on Spontaneous GABAergic Synaptic Activity in CA1 Pyramidal Cells

To investigate the effect of LTG on inhibitory transmission onto pyramidal cells in CA1, whole-cell voltage-clamp recordings from neurons were made under control conditions and after bath application of LTG (50 μM). Cells were clamped at 0 mV to record the spontaneous inhibitory postsynaptic currents (sIPSCs). The representative traces of sIPSCs in the absence or presence of LTG (50 μM) are shown in Figure 2. Compared to the control, LTG had negligible effect on the mean value of sIPSC amplitude (baseline: 19.7 ± 1.19 pA; LTG: 18.96 ± 0.93 pA; number of slices = 7, *p* = 0.26, paired *t*-test) or frequency (baseline: 1.91 ± 0.34, LTG: 1.94 ± 0.58 Hz, number of slices = 7, *p* = 0.92, paired *t*-test). Clearly, LTG did not affect GABAergic transmission onto pyramidal cells in CA1 region, indicating neuron specific effects on the different pyramidal neuron circuits.

### 2.3. GABAergic Transmission Differs between CA1 and CA3c Hippocampal Neurons

The CA1 and CA3c regions of HPC neuronal networks participate in different extended circuitries [43,44] and hence, the inhibitory input of these areas may vary and differentially regulate the local network. To gain insight into the inhibitory regulation of local CA1 and CA3 networks, we examined the characteristic of the sIPSCs in pyramidal CA1 and CA3 neurons. The frequency of the sIPSCs was higher (7.05 ± 1.48 Hz) in the CA3c as compared to the CA1 (1.91 ± 0.34 Hz, *p* = 0.0003, *t*-test, Figure 3), whereas the amplitudes of the currents differed, but to a lesser extend (CA3c: 35.24 ± 4.03 pA, CA1: 19.7 ± 1.19 pA, *p* = 0.04, Wilcoxon Signed Rank Test). Further analysis of the sIPSC kinetic parameters showed that the mean rise times and the mean decay times of the sIPSCs from the two regions were similar (Figure 3). Taken together, these results suggest a significant difference in functional GABA_A_-mediated fast inhibition between CA1 and CA3.

### 2.4. Effects of LTG on CA1 Pyramidal Neuron Excitability and Passive Membrane Properties

Similar to effects seen on interneurons, application of LTG significantly attenuated the excitability of CA1 pyramidal cells, as indicated by a flatter action potential frequency-current relationship and lower total number of action potentials generated per cell (106 ± 14 vs. 76 ± 12, *t* = 5.4, df = 10, *p* < 0.001, paired *t*-test, Figure 4A; Top). Furthermore, LTG significantly depolarized the resting membrane potential (−65.77 ± 0.94 mV vs. −63.48 ± 1.04 mV, *t* = −4.54, df = 10, *p* = 0.0011, paired *t*-test; Figure 4B; Top), but no significant changes in cell input resistance was observed, although a tendency to decrease was detected (102.21 ± 12.16 MΩ vs. 84.53 ± 7.48 MΩ, *t* = 2, df = 10, *p* = 0.073, paired *t*-test; Figure 4C; Top). In the presence of the selective HCN channel blocker ZD 7288 (10 µM), LTG still decreased neuronal excitability, resulting in a lower total number of action potentials generated in response to a series of current pulses (151 ± 17 vs. 129 ± 16, *t* = 3.58, df = 7, *p* = 0.009, paired *t*-test), as shown in Figure 4A; Bottom. Thus, its mechanism of action on CA1 neuronal excitability is at least partially independent of HCN channel modulation. However, ZD 7288 prevented LTG-induced depolarization (−75.49 ± 1.38 mV vs. −75.51 ± 1.86 mV, *t* = 0.02, df = 7, *p* = 0.99, paired *t*-test; Figure 4B; Bottom). The change in input resistance also seemed to be reversed, with an overall increase despite the lack of statistical significance (228.12 ± 22.18 MΩ vs. 268.75 ± 14.35 MΩ, *t* = −1.76, df = 7, *p* = 0.12, paired *t*-test; Figure 4C; Bottom). Therefore, modulation of passive membrane properties by LTG depends on HCN channels, but effects on other channels (Na/Ca) contributing to depressed excitability.

### 2.5. Effects of LTG on the Resonant Properties of CA1 Neurons

The resonant frequency of the cells did not change significantly following LTG treatment (3.28 ± 0.17 Hz vs. 3.28 ± 0.19 Hz, *t* = 0, df = 9, *p* = 1, paired *t*-test, Figure 5B). Similarly, no changes in the resonant impedance were observed (109.36 ± 7.4 MΩ vs. 103.6 ± 9.25 MΩ, *t* = 0.88, df = 9, *p* = 0.4, paired *t*-test, Figure 5C).

### 2.6. Effect of LTG, ZD 7288, and the Combined Infusion on Cholinergically Induced Hippocampal Theta Oscillations In Vitro

The effects of different concentrations of LTG, ZD 7288, and the combination of LTG and ZD 7288 on LFPs in acute HPC slice preparations were studied. Field theta oscillations (control recordings) were evoked by the application of the cholinergic agonist carbachol (50 μM), which is necessary to induce well-synchronized theta epochs in vitro corresponding to those recorded in vivo [45,46]. When sequentially perfused with LTG at a concentration of 30 μM (*n* = 28 slices) or 50 μM (*n* = 32 slices) and coapplied with 50 μM carbachol, the HPC slices responded with no changes in recorded theta oscillations (data not shown). When perfused with a combination of 100 μM LTG and 50 μM carbachol (*n* = 45 slices), acute HPC slices responded with a complete decay of theta epochs at 2–3 min post application (Figure 6A). Application of ZD 7288 at a concentration of 10 μM in combination with 50 μM carbachol (*n* = 38 slices) resulted in complete electrophysiological “silencing” at 2–4 min post application. Combined application of 100 μM LTG and 10 μM ZD 7288 together with 50 μM carbachol (*n* = 40) completely abolished cholinergically induced theta oscillations. After each recording obtained from CA3c, LFPs were also recorded in the CA1 layer; however, they did not differ from CA3c, except for an overall lower amplitude compared to CA3c (data not shown).

### 2.7. Effect of LTG, ZD 7288, and a Combined Infusion on Spontaneous Hippocampal THETA Rhythm Recorded In Vivo

Four experimental groups (groups I, II, II, IV, *n* = 5 rats/group) were established, depending on the dose and combination of agents used in the protocol (Table 1). It should be noted that recordings were made using urethane anesthesia, so only effects on type 2 (cholinergic) theta were measured [47]. Urethane is known to attenuate excitability, but otherwise leaves theta unchanged [48]. The effects induced by the intra HPC injection of LTG (4 μg/μL) are shown in Figure 6. Furthermore, the effects of a lower dose of LTG (3 μg/μL; *n* = 5) on spontaneous theta oscillations were studied; but did not produce any changes in theta rhythm parameters (data not shown). Figure 6 provides a representative example of recording from one animal, illustrating the effect of LTG (4 μg/μL) on spontaneous HPC type II theta rhythm in anaesthetized rats along with the corresponding spectrogram estimated from each data segment at pre- and postinjection times. The details concerning changes in the average theta frequency, amplitude, and power at pre- and postinjection time points (30, 60 and 120 min) are presented in Figure 7 and in Table 1. At 30 and 60 min after the LTG (4 μg/μL) injection, the power and amplitude increased substantially, with no differences at 120 min postinjection (Figure 7A and Table 1). Intra HPC LTG (4 μg/μL) administration did not induce any significant changes in theta frequency within 30, 60 and 120 min of recording (Figure 7A and Table 1). When LTG was applied at 6 μg/μL, we observed a significant decrease in power and amplitude but no change in frequency (Table 1). A specific antagonist of HCN channels ZD 7288 applied at a dose of 4 μg/μL (group III; *n* = 5) led to a very significant reduction in the power and amplitude of theta oscillations, with no effect on frequency (Figure 7B and Table 1). Application of the combination of LTG and ZD 7288 was used (group IV, *n* = 5) to verify whether the observed effect of LTG was due to selective interaction with HCN channels. Intra HPC injection of the combination of 4 μg/μL ZD 7288 and 4 μg/μL LTG did not alter spontaneous theta power, amplitude, and frequency at 30, 60 and 120 min of recording (Figure 7C and Table 1). The control intra HPC vehicle and sham injections did not induce any changes in HPC local field potentials (LFPs) at 30, 60 and 120 min of recording in experimental groups I, II, II and IV (data not shown).

## 3. Discussion

Our results agree with previous studies demonstrating LTG depressed pyramidal cell excitability and reduced synaptic inhibition in the HPC. Depressed synaptic inhibition appears to be secondary to depressed excitability of inhibitory interneurons. We also show that effects on synaptic inhibition were cell-type specific, which can explain earlier conflicting reports of LTG’s effects. Depressed excitability of pyramidal neurons readily accounts for the observed depression of theta frequency LFP oscillations in brain slices and likely comes about through enhanced HCN channel activity and depressed cation currents. The enhancing effect of LTG on theta seen in vivo indicates that ascending oscillatory inputs overpower local depressant effects produced in HPC. Depressed pyramidal cell excitability and enhanced theta activity could combine to produce LTG’s antiepileptic actions.

### 3.1. LTG Reduces Spontaneous GABAergic Synaptic Activity in CA3c, with No Effect on CA1

#### Pyramidal Cells

Our in vitro whole-cell voltage-clamp recordings in CA1 and CA3c areas of acute HPC slices revealed that LTG application selectively decreased the frequency and amplitude of sIPSCs on CA3c pyramidal cells. The decreased frequency might result from reduced firing of presynaptic cells and/or decreased neurotransmitter release, whereas the reduced amplitude might be determined by pre- or postsynaptic factors. Moreover, the reduction in sIPSC frequency and amplitude observed in CA3 was absent when TTX was applied, suggesting that this effect was mediated by depressed interneuron discharge. At the same time, we did not observe any consistent differences in these properties in CA1 pyramidal cells. Regional differences in GABAergic innervation of CA1 vs. CA3 [43,44,49] as well as HCN subunit expression patterns [50,51,52], may account for this region-specific reduction in the sIPSC frequency and amplitude only in CA3c pyramidal cells. Regional differences in entorhinal innervation and network connectivity could also contribute because these inputs are exclusive of CA3 [53,54]. Differences in the expression patterns, subcellular localization, and co-expression of HCN subunits are responsible for the different biophysical properties of HCN channels and therefore might partially contribute to the different physiological roles of I*h* in various brain regions [12]. For example, the HCN1 subunit is expressed at the highest levels in adult HPC CA1 neurons, which exhibit a very rapidly activating I*h* [14]. This property may be contrasted with adult HPC CA3 pyramidal cells, where HCN1 is not detectable [27,52], while HCN2 isoform expression is evident in CA1 and CA3 at both the mRNA and protein levels [51]. Interestingly, two independent studies that focused on the effects of LTG on the sIPSC frequency in CA1 reported contradictory results [37,55]. A possible explanation for this inconsistent finding is the different ages of rats, suggesting developmental changes [56,57]. Alternatively, a possible explanation for the lack of effect on sIPSC frequency onto CA1 pyramidal cells is that our recordings were obtained from coronal HPC slices, while previous studies were carried out on transverse HPC slices, also we used a lower dose of LTG (50 μM vs. 100 μM).

### 3.2. LTG Attenuates Membrane Excitability and Depolarizes the Resting Membrane Potential of CA1 Pyramidal Neurons with No Effect on Membrane Resonant Properties

Whole-cell current clamp recordings from CA1 pyramidal cells revealed that LTG decreased neuronal excitability while simultaneously depolarizing the resting membrane potential. Here, blocking HCN channels with ZD 7288 did not completely prevent the LTG-mediated decrease in excitability, indicating that another ionic conductance was involved. LTG is a known modulator of voltage-dependent Na^+^ and Ca^2+^ channels [58,59]. LTG-induced depolarization of the resting membrane potential was abolished in slices pretreated with ZD 7288, indicating that the effect was mediated by the activation of HCN channels. Nonsignificant but noticeable changes in the cell input resistance were produced by both LTG and ZD 7288, consistent with HCN channel involvement. Our results are consistent with previous findings, as the opening of HCN channels is known to lower membrane resistance and depolarize membrane potentials [12]. The reversed tendency of the input resistance to increase after LTG treatment preceded by blocking HCN channels might be explained by the unmasking of a second ionic mechanism, such as the closure of voltage-dependent Ca^2+^ channels.

HCN channels are known modulators of theta oscillations partly due to their ability to tune the membrane potential resonance properties to the theta frequency band [60]. Therefore, we wanted to determine whether LTG would affect the membrane potential resonance in CA1 pyramidal cells by examining the 1–20 Hz impedance profile calculated based on the membrane voltage response to chirp waveform stimulation. Surprisingly, LTG did not change either the resonant frequency or the resonant impedance of the neuronal membrane.

We show that LTG increases the amplitude and power of spontaneous HPC theta rhythms recorded from the CA1 region in vivo at a concentration of 4 μg/μL, whereas ZD 7288 significantly decreases measured theta parameters. In contrast to this observation, the application of LTG in vitro completely inhibits theta oscillations. Moreover, our results indicate that LTG reduces spontaneous GABA synaptic activity in CA3c pyramidal cells with no effect on GABAergic transmission in the CA1 subfield. Additionally, the application of LTG significantly attenuates the excitability and depolarizes the resting membrane potential of CA1 pyramidal cells. Hippocampal theta oscillations have different characteristics in vivo and in vitro. Both share very similar cholinergic and GABAergic profiles [61,62]; however, theta oscillations recorded in vivo are strongly influenced by external inputs from regions such as the medial septum and the diagonal band of Broca (MS/DBB) [63,64,65], as well as the entorhinal cortex [3,66]. Our previous studies of the laminar profile of HPC theta oscillations in vitro and in vivo clearly revealed that theta oscillations generated in CA3c-transected slices are characterized by a higher amplitude and higher level of synchrony than CA1 theta oscillations [67]. At the same time, HPC theta oscillations observed in vivo in CA1 are driven by both intra HPC (CA3c) and extra HPC inputs [3,66]. Furthermore, in contrast to CA1 pyramidal neurons, their CA3c counterparts are more prone to fire spontaneous bursts of action potentials [68], which in turn may manifest in their easier disinhibition caused by decreased GABAergic tone after LTG treatment, as shown in our work.

### 3.3. LTG at a Concentration of 4 μg/μL Enhances Hippocampal Theta Oscillations In Vivo

Selective agonists are unavailable for HCN channels; therefore, LTG is commonly used to study HCN function. Although LTG is also known to inhibit Na^+^ and Ca^2+^ channels, it acts mainly on the slow inactivated state with no effect on the resting closed state [58,59]. Furthermore, the LTG-induced decrease in membrane excitability was not associated with a change in action potential parameters and was blocked by prior application of the highly specific HCN antagonist ZD 7288, suggesting that LTG mainly acts through I*h* channels to depress neuron discharge [35]. Additionally, LTG enhances the I*h* mediated by HCN channels in pyramidal neurons through a positive shift in the voltage dependence of I*h* activation [31]. I*h* regulates membrane properties involved in rhythmic firing, subthreshold MPOs, and dendritic integration [69,70]. According to previous studies, I*h* may function as a pacemaker current in the entorhinal cortex and acute HPC slices [71,72]. The close relation between MPOs and extracellular HPC theta oscillations has been discussed previously in detail [10,73]. In summary, MPOs and rhythmic cell discharges are phase-locked with extracellular recorded theta field potentials [10]. We did not record the activity of single HPC neurons in the presence of LTG in this study; however, the fact that theta field activity was repeatedly facilitated in response to the administration of LTG in vivo indirectly suggests a temporal increase in the number of active HPC theta-on phasic cells during theta rhythm induced by this agent. In contrast to the increased amplitude and power of HPC theta rhythm, we did not observe any changes in terms of the frequency of spontaneous theta oscillations following LTG administration. These data provide additional evidence supporting an earlier hypothesis that the rat HPC has a unique contribution to programming only theta amplitude but not frequency [2,63]. Our results additionally indicate that the HPC neural mechanism responsible for the programming of theta amplitude and power in vivo may also involve I*h*. Moreover, we observed a dose-response effect, suggesting that effective enhancement of theta oscillations may occur in vivo only when LTG is applied at a strictly defined concentration (4 μg/μL). Although experimental data from similar studies are currently lacking, a computational study has investigated the role of HCN channels in regulating LFPs and the theta-frequency spike phase. This experiment was based on 440 neuropil-containing, morphologically realistic, conductance-based CA1 pyramidal neuronal models [74]. The authors describe an ability of HCN channels to introduce an inductive phase leading to intracellular voltage responses to theta-modulated synaptic currents, which played an important role in altering LFPs and spiking. Other authors have shown that CA1 pyramidal neurons use a gradient of inductance through HCN channels as an active mechanism to counteract location-dependent temporal differences in dendritic inputs at the soma. Using simultaneous multisite whole-cell recordings and computational modelling, Vaidya and Johnston [75] found that this intrinsic biophysical mechanism produces temporal synchrony of rhythmic inputs in the theta frequency across the dendritic tree, which in turn coincide with the appearance of extracellularly recorded theta oscillations. The involvement of I*h* in rhythmogenesis has been described since the early 1980s in thalamocortical relay neurons [76]. The ionic requirements for electroresponsiveness in these neurons were studied using rodent slice preparations in vitro. The biophysical properties of thalamic neurons allow them to serve as single-cell oscillators at two distinct frequencies, 9–10 Hz and 5–6 Hz, coinciding with alpha and theta rhythms [76]. Later, Kocsis and Li [13] investigated the role of Ih in rhythmogenesis in vivo. They found that blocking I*h* in the medial septum substantially decreased the frequency of HPC oscillations without changing the context in which theta oscillations occurred, i.e., specific behaviours in freely moving rats. Thus, functional I*h* was necessary for the septum to generate or transmit high-frequency theta rhythms elicited by strong ascending activation. Based on these results, I*h* plays a specific role in septal theta generation by promoting theta rhythms during exploratory behaviour and rapid eye movement sleep. This result might partially explain our findings by suggesting a substantial role for I*h* in modulating theta amplitude and power in vivo. Sotty et al. [77] found that I*h* conductance facilitated burst firing, shortened the delay of rebound-firing hyperpolarization, and increased its temporary frequency, inducing resonance of cells and allowing rhythmic input transmission in a relatively wide theta range. Hu et al. [78] showed that I*h* contributes to theta resonance at membrane potentials negative to −80 mV in CA1 pyramidal cells. Theta resonance is also present in specific interneurons in the stratum oriens of the HPC, which are also rich in HCN channels [79]. Sotty et al. [77] proposed that I*h* present in medial septal neurons more likely supports resonance than pacemaker activity since the presence of I*h* allowed the modulation of theta frequency in tune with the level of ascending drive from the brainstem. Moreover, in the MS/DBB, which is the upper theta rhythm generator, I*h* was detected in bursting GABAergic neurons [13,77,80,81]. In 2004, Xu et al. [82] showed that selective blockade of HCN channels by ZD 7288 reduced the spontaneous firing of septo-HPC GABAergic neurons in rat brain slices, and local infusions of ZD 7288 into the MS/DBB reduced exploration and sensory-evoked HPC theta bursts in behaving rats. Therefore, they proposed that I*h* in septo-HPC GABAergic neurons modulates the HPC theta rhythm. ZD 7288 also attenuates excitability in neocortical pyramidal neurons [83] and decreases neocortical hyperexcitability by depressing synaptic transmission [84].

Interestingly, our data revealed that direct, intra HPC injections of LTG only at a dose of 4 μg/μL significantly increased the theta amplitude and power in vivo, while ZD 7288 significantly decreased those parameters in vivo. Consistent with this finding, Nolan et al. [25] documented the role of HCN1 channels in the development of spatial memory and plasticity in CA1 pyramidal neurons using mice with general or forebrain-restricted knockout of the HCN1 gene. Later, the same authors suggested that HCN1 channels expressed by stellate neurons in layer II of the entorhinal cortex are key molecular components in the processing of inputs to the HPC DG, with distinct integrative roles during resting and active states [85].

### 3.4. LTG Diminishes Hippocampal Theta Oscillations In Vitro

Hippocampal acute slices responded with a complete disappearance of theta epochs recorded from the CA3c region a few minutes after the application of LTG, as well as ZD 7288. We should emphasize that our in vitro recordings were always obtained from the CA1 subfield apart from CA3 recordings; however, theta parameters recorded from CA1 were worse than those in the CA3 region (data not shown). As mentioned above, the mechanism of action of LTG, in addition to enhancing I*h*, is to block voltage-gated Na^+^ and Ca^2+^ channels and inhibit activity-driven glutamate release from presynaptic terminals of excitatory neurons [58,59]. These actions most likely depressed theta oscillations in completely isolated HPC slices, especially when used at relatively high concentrations. LTG (30–100 μM) that reduces glutamate release in DG granule cells and inhibits both EPSC_AMPA_ and EPSC_NMDA_ evoked responses [86]. In addition to cholinergic and GABAergic transmission, glutamatergic drive is also important for the appearance of HPC theta oscillations, especially in vitro [87]. LTG appears to cause an imbalance in the overall level of network excitation and inhibits spontaneous activity of pyramidal CA3c and CA1 neurons necessary for theta oscillations to appear in vitro. Interestingly, Adams et al. [88] observed a reduction (10%) in the burst duration of CA1 pyramidal neurons after LTG treatment, which was associated with a reduction in voltage-dependent Ca^2+^ current. In HPC neurons, HCN channels preferentially target distal dendrites, where they dampen dendritic excitability and dendritic temporal integration and reduce the EPSP amplitude [12,89,90,91]. At the same time, GABAergic transmission is vital for the appearance of HPC theta oscillations in vitro [62,87].

### 3.5. Summary Conclusions

Neuronal populations in acute HPC slices are completely devoid of ascending influence from the MS/DBB and entorhinal cortex, which are critical for the development of theta oscillations in the dorsal CA1 [3,63,65,66,86]. The LTG-mediated enhancement of HPC theta oscillations requires preserved inputs from the septo-HPC pathway as well as entorhinal cortex. Our patch clamp recordings revealed that LTG decreased sIPSC frequency and amplitude recorded in CA3c pyramidal neurons and indicated depressed interneuron discharge in the CA3c region, with no changes in local interneuron activity in the CA1 region. Therefore, network activity is at least somewhat sensitive to LTG treatment, consistent with a network hypothesis for LTG’s antiepileptic actions, i.e., increased theta = better anticonvulsant effects [92,93]. Our paper provides a better understanding of how lamotrigine modulates the intrinsic and extrinsic electrophysiological properties of HPC neurons, dependent on dosage, cell type and sub-region.

## 4. Materials and Methods

The studies described below were approved and monitored by the Local Ethics Committee for Animal Experiments in Lodz (Permissions Number: 41/ŁB67/2017). All surgeries were performed under anaesthesia, and all efforts were made to minimize animals’ suffering.

### 4.1. Patch Clamp Electrophysiology

#### 4.1.1. Tissue Preparation for Whole-Cell Patch-Clamp Recordings

Male Wistar (4–7-week-old) rats (25 animals) were anaesthetized with isoflurane (Aerrane, Baxter, Warsaw, Poland) and decapitated. Brains were quickly removed and placed in ice-cold modified ACSF (artificial cerebrospinal fluid) containing (in mM) 130 NaCl, 5 KCl, 2.5 CaCl_2_, 1.3 MgSO_4_, 1.25 KH_2_PO_4_, 26 NaHCO_3_, and 10 D-glucose (bubbled with 95% O_2_/5% CO_2_), pH 7.4; osmolality 290–300 mOsmol kg^−1^). Coronal sections (300 µm thick) were cut using a vibrating microtome (VT1000, Leica Biosystems, Nussloch, Germany). Slices containing the HPC were transferred and held in an incubation chamber filled with carbogenated ACSF at 32 °C for at least 1 h. After this recovery period, individual slices were transferred to a recording chamber placed on a fixed stage of a Zeiss AxioExaminer A1 (Carl Zeiss Microscopy, Jena, Germany) upright microscope, where the tissue was perfused (2–3 mL/min) with carbogenated, warm (32 °C) ACSF containing (in mM): 132 NaCl, 2 KCl, 1.25 KH_2_PO_4_, 26 NaHCO_3_, 2.5 CaCl_2_, 1.3 MgSO_4_ and 10 glucose, pH = 7.35; 290–300 mOsmol kg^−1^).

#### 4.1.2. Whole-Cell Patch-Clamp Recordings

Neurons were identified based on their location and somatic morphological characteristics under an infrared differential interference contrast (DIC) microscope. The image from the microscope was enhanced using a CCD camera and displayed on a computer monitor. The border of CA1 or CA3c was identified based on an image captured with a low magnification objective (2.5×) and referenced to the Paxinos Atlas [94]. Neurons with a healthy appearance presented a smooth surface, and the cell body and parts of the dendrites were clearly visible. Neurons were approached under visual control with a patch pipette using a three-axis micromanipulator (uMp micromanipulator system, Sensapex, Oulu, Finland). Recording micropipettes were fabricated from borosilicate glass capillaries (3–6 MΩ) using a horizontal puller (P-97, Sutter Instruments, Novato, CA, USA) and filled with the following solutions: in experiments measuring inhibitory synaptic transmission, the solution contained, in mM, 122 Cs-gluconate, 10 CsCl, 10 HEPES, 0.3 EGTA, 10 phosphocreatine-Na_2_, 5 Mg-ATP, and 0.4 Na-GTP, pH = 7.35; osmolality 280–290 mOsmol kg^−1^. In current clamp experiments, the intracellular solution contained, in mM, 122 K-gluconate, 10 KCl, 10 HEPES, 0.3 EGTA, phosphocreatine-Na_2_, 5 Mg-ATP, and 0.4 Na-GTP, pH = 7.35; osmolality 280–290 mOsmol kg^−1^. Whole-cell recordings were obtained from CA1 or CA3c pyramidal neurons, identified based on their response to hyper- and depolarizing current pulses. After break-in, the cells were stabilized for 5 min before commencing recordings. Signals were recorded using an Axopatch 1D (Axon Instruments, Novato, CA, USA) for voltage-clamp recordings or a Multiclamp 700B amplifier (Molecular Devices, San Jose, CA, USA) for current-clamp recordings, filtered at 2 kHz and 10 kHz for voltage- and current-clamp recordings, respectively, and digitized at 20 kHz and 50 kHz using the Digidata 1440 interface and pClamp 10 software (Molecular Devices, San Jose, CA, USA). Voltages were corrected for the liquid junction potential (experimentally measured as −12 mV). Access resistance was monitored throughout each experiment. Only recordings with stable access resistance lower than 30 MΩ were considered acceptable for analysis. All drugs were delivered via a bath perfusion system.

#### 4.1.3. Recording and Detection of Spontaneous Inhibitory Postsynaptic Currents

Neurons were voltage-clamped at 0 mV to record spontaneous inhibitory postsynaptic currents. After 15 min of stabilization, the baseline sIPSC activity was recorded. The effects of LTG administration (50 μM) were assessed by recording the slices 10 min before (baseline) and during the 15 min of LTG superfusion. In a subset of experiments, slices were incubated in ACSF supplemented with an HCN channel-selective antagonist, ZD 7288 (10 μM), to verify whether the effect of LTG was HCN-dependent. Spontaneous IPSCs were detected using the TaroTools toolbox for Igor Pro (https://sites.google.com/site/tarotoolsregister, accessed on 12 January 2021).

#### 4.1.4. Whole-Cell Current-Clamp Recordings of CA1 Pyramidal Neurons

##### Synaptic Blockers

The following synaptic blockers were included in ACSF to effectively isolate single neurons from the effect of network activity: NBQX (5 µM, AMPA/kainate antagonist), CGP 37849 (10 µM, NMDA receptor antagonist), and bicuculline methiodide (5 µM, GABA_A_ receptor antagonist). These compounds were all purchased from Tocris, Bristol, UK, and were used throughout all current-clamp recordings.

##### Excitability and Passive Membrane Properties

After successful patching followed by a 10 min stabilization period, the resting membrane potential (RMP) of each recorded neuron was recorded for 30 s in current-zero mode (and calculated as the mean value of the membrane potential for this recording duration). Next, neurons were maintained in a current clamp at −70 mV (taking into account the −12 mV liquid-liquid junction potential) for the measurement of the frequency-current relationship. For this experiment, each cell was subjected to a series of hyper- and depolarizing 500 ms current steps from −60 pA to +320 pA in 20 pA increments. Following baseline measurements, 50 µM LTG (Tocris, Bristol, UK) was added to the ACSF. After 15 min, the current steps were repeated to measure changes after treatment. Recordings were continuously monitored for changes in access resistance, and only recordings with stable access resistance of less than 30 MΩ throughout the recording were accepted for analysis.

##### Membrane Potential Resonance

The cell was in current-clamp mode, and the neuronal membrane potential was held at approximately −60 mV to test whether the application of LTG affected intrinsic resonance properties. Cells were subjected to a 20 s chirp waveform current command, i.e., a cosine with constant amplitude and linearly increasing frequency from 1–20 Hz. The resulting voltage response had a characteristic “bump,” indicative of resonant properties of the cell membrane (see Figure 3A). The impedance profile of the cell membrane for a particular neuron was obtained by plotting the calculated FFT of the voltage response divided by the FFT of the current stimulus waveform. The resonant frequency was defined as the frequency at which the smoothed impedance profile reached the maximum value. The resonant impedance was defined as the membrane impedance at the resonant frequency.

#### 4.1.5. Patch Clamp Data Analysis and Statistics

For sIPSC recordings, amplitude detection thresholds were set to exceed noise values. Events were detected automatically, analysed for double peaks, and then visually inspected to prevent the inclusion of false-positive data. The synaptic activity was analysed for frequency, amplitude, rise time (calculated from 10–90% of the peak amplitude), and decay time calculated by fitting an exponential function from 10–90% of the decay phase. All statistical tests and plotting were performed using R version 3.6.1 [95] and the dplyr version 0.8.3 [96] and ggplot2 version 3.2.1 packages [97]. Group data for inhibitory synaptic transmission are reported as the medians ± SD. The effect of lamotrigine on sIPSC characteristics was analysed by comparing pre- and post-LTG values using a paired Student’s *t*-test or the Wilcoxon signed rank test for nonnormally distributed data.

For current-clamp recordings, the number of action potentials generated in response to each current step was determined, along with the input resistance (calculated from the mean voltage change in response to the −20 pA and +20 pA current pulses). Data analysis, generation of plots and statistics were performed using R language version 4.0.5 [95] and the ggplot2 and dplyr packages [97,98]. The mean values for the number of action potentials generated by each current pulse before and after the application of LTG were fitted with a 5-parameter logistic equation (sigmoidal function), with the “bottom” parameter shared between the baseline and LTG conditions, using the R package drc [99]. The total number of action potentials generated by a particular cell throughout the current step protocol was also determined and compared before and after LTG application. Mean values obtained before and after LTG administration were compared using appropriate statistical tests indicated in the Results section. The membrane potential resonance analysis and construction of impedance profiles were performed in IgorPro 8 (WaveMetrics, Lake Oswego, OR, USA). Unless indicated otherwise, current-clamp data are presented as the means ± SEM.

### 4.2. In Vitro Local Field Potentials (LFPs) Recordings

#### 4.2.1. Subjects and Procedure

Experiments were performed on 183 HPC slices obtained from 35 male Wistar rats (~100–150 g). Each animal was first anaesthetized with isoflurane (Aerrane, Baxter, Warsaw, Poland) and then decapitated. The brain was rapidly removed and placed in oxygenated (95% O_2_ + 5% CO_2_), cold (~5 °C) artificial cerebrospinal fluid (ACSF; composition in mM: 121 NaCl, 5 KCl, 2.5 CaCl_2_, 1.25 KH_2_PO_4_, 1.3 MgSO_4_, 26 NaHCO_3_, and 10 glucose; Sigma-Aldrich, St. Louis, MO, USA), which was freshly prepared before each experiment using prefiltered and deionized water (Easy Pure RF, Waltham, MA, USA). The HPC was extracted separately from the left and right hemispheres and then cut into slices (~500 µm thick) in the coronal plane using a tissue slicer (Stoelting, Wood Dale, IL, USA). Slices were then preincubated in oxygenated ACSF at room temperature for 1 h after dissection. Next, the slices were transferred into the gas-liquid interface recording chamber and maintained on a nylon mesh for 0.5 h before the recording. Afterwards, slices were continuously perfused with oxygenated and prewarmed (35 °C) ACSF at a flow rate of approximately 1.5 mL/min.

#### 4.2.2. Recording Technique, Data Acquisition and Data Analysis

Local field potentials in all HPC slices were induced by perfusion with carbachol (carbamoylcholine chloride—CCH; Sigma-Aldrich, St. Louis, MO, USA) dissolved in ACSF to a concentration of 50 µM. In vitro experiments were divided into three experimental groups (each containing 61 HPC slices). After the induction of well-synchronized theta oscillations in HPC slices, different agents (Sigma-Aldrich, St. Louis, MO, USA) were applied to examine their effects on CCH-induced theta rhythms: (1) ZD 7288 (10 µM), (2) LTG (100 µM), and (3) ZD 7288 (10 µM) + LTG (100 µM). All chemicals were dissolved in ACSF containing CCH (50 µM). Glass recording electrodes (5–9 MΩ) filled with 2.0 M sodium chloride were pulled from Kwik-Fill capillaries (World Precision Instruments, Sarasota, FL, USA). The recording electrode was positioned in the CA3c region of the HPC slice using a single-axis motorized micromanipulator (IVM-1000, Scientifica, Uckfield, UK). The LFPs recorded with respect to ground. The preamplified signal was fed into an AC amplifier (model p-511, Grass-Astromed, West Warwick, RI, USA) with the high-pass filter set at 1 Hz and the low-pass filter set at 0.3 kHz. All signals were digitized by an analogue-to-digital converter (Power 1401 ADC, Cambridge Electronic Design, Cambridge, UK). The field recording trace was displayed online on a PC monitor through Spike 2.7 software (Cambridge Electronic Design, Cambridge, UK). Recordings were stored on a PC hard drive for subsequent off-line analysis with Spike 2.7 software.

### 4.3. In Vivo Local Field Potentials (LFPs) Recordings

#### 4.3.1. Subjects and Surgical Procedure

The data were obtained from 35 Wistar rats (200–250 g) housed on a 12 h light/dark cycle with free access to water and food. The rats were initially anaesthetized with halothane (Sigma-Aldrich, St. Louis, MO, USA), and a jugular cannula was inserted. Halothane was then discontinued, and urethane (0.6 g/mL, Sigma-Aldrich, St. Louis, MO, USA) was administered via the jugular cannula to maintain anaesthesia throughout the experiment. The anaesthesia level was maintained to ensure that theta field potentials and the transition from theta to large irregular activity would occur spontaneously. Body temperature was monitored and maintained at 36.5 ± 0.5 °C with a heating pad, and heart rate was monitored constantly throughout the experiment.

#### 4.3.2. Hippocampal Electrode Implantation and Local Field Potential Recording

Rats were placed in a stereotaxic frame, and the plane between the bregma and lambda was adjusted to horizontal. An uninsulated tungsten wire placed in the cortex 2 mm anterior to bregma served as a reference electrode, and the stereotaxic frame was connected to the ground. A tungsten microelectrode (0.1–0.9 MΩ) for recording HPC local field potentials was placed in the right dorsal HPC in the stratum lacunosum-moleculare (3.7 mm posterior to the bregma, 2.0–2.2 mm lateral from the midline and 2.4–2.6 mm ventral to the dural surface) [94]. During experiments, AC amplifiers (P-511, Grass-Astromed, West Warwick, RI, USA) were used for recording HPC field potentials, with high-pass and low-pass filters set to 1 Hz and 0.3 kHz, respectively. The field activity was displayed using a digital storage oscilloscope (TDS 3014B; Tektronix, Beaverton, OR, USA). Signals were digitized by the Micro 1401 interface (Cambridge Electronic Design, Cambrige, UK) and saved onto a computer hard drive for subsequent off-line analysis (Spike 2.7 Cambridge Electronic Design, Cambridge, UK).

#### 4.3.3. Hippocampal Cannula Implantation and Injections

Microinjections (26 gauge, 5 mL Hamilton 701N syringe) of pharmacological agents were always performed in the right HPC at a rate of 0.5 μL/60 s. All drugs used in this study were obtained from Sigma-Aldrich Chemical Corporation (St. Louis, MO, USA). The coordinates for the Hamilton cannula were as follows: 4.3 mm superior to the bregma, 2.6 mm lateral from the midline, and 2.4–2.6 mm ventral to the dural surface [94].

#### 4.3.4. Experimental Procedure

Two different pharmacological agents were used in separate experiments to characterize the effect on HPC spontaneous theta rhythm. In the first step of the experiments, three doses (3, 4 and 6 μg/μL) of the HCN channel opener LTG were tested (5 rats per dose). In the second step, three doses (2, 3 and 4 μg/μL) of a specific HCN channel blocker (ZD 7288) were tested (5 rats per dose). In the last stage, LTG was injected into the HPC after a previous local injection of ZD 7288. All agents were dissolved in pharmacological saline (Polpharma, Warsaw, Poland).

Five minutes of control recordings of HPC spontaneous LFPs were obtained prior to each microinjection to characterize the effect of LTG and ZD 7288 on HPC theta oscillations. After the administration of LTG and/or ZD 7288, another 5 min recordings were obtained at successive postinjection time periods (30, 60 and 120 min). Subsequently, 3 s samples containing theta rhythms were selected for analysis from successive time periods.

#### 4.3.5. Recording Procedure and Data Analysis

The LFPs were recorded continuously and analysed in 5 min panels before drug injection (control) and at strictly defined time points after the intra HPC drug injection (30, 60, and 120 min). From each panel, ten 3 s samples of theta rhythms were selected for analysis. A theta epoch was recognized as rhythmic high-amplitude sinusoidal waveforms in a defined frequency band (3–6 Hz). The following parameters of HPC theta field activity were analysed: power, amplitude and frequency. The power and frequency were determined using the fast Fourier transform (FFT, 128 size, Hanning window, frequency from 0 Hz to 50 Hz in 64 bins, resolution 0.7813 Hz). The peak power in the theta frequency band was directly obtained from the peak of the FFT, along with the frequency of the dominant power within the range of 3–6 Hz. The mean peak-to-peak amplitude of theta oscillations was determined directly from the theta epochs. Power spectra in the 0–50 Hz range for the HPC field potential recordings (obtained at a sample rate of 100 Hz) were generated using the FFT algorithm implemented in the Spike 2.7 software package (Cambridge Electronic Design, Cambridge, UK). The spectrograms (top dB 96 × pixel 1, Hanning window, range dB 96, block size 1024) were taken from 30 s epochs of the HPC theta field potentials.

#### 4.3.6. Histological Procedure

Following data acquisition, rats were sacrificed by an overdose of urethane for the histological examination. The brain was removed, stored in 10% formalin, frozen and subsequently sectioned into 20 µm slices in the sagittal plane. Next, the slices were mounted on glass slides for subsequent verification of the HPC electrode and injection cannula placement.

#### 4.3.7. Statistics

The statistical calculations were performed with STATISTICA software (version 13 1; TIBCO Software, Palo Alto, CA, USA). The sample size was estimated for type I and type II statistical errors of 0.05 and 0.8, respectively. The results are presented as the medians and interquartile ranges (lower-upper quartiles, 25–75%), *n* = 5 animals per group. The data were tested for a normal distribution with the Shapiro-Wilk test, and the homogeneity of variance was verified with the Brown-Forsythe test. The statistical significance of differences between groups was evaluated using repeated measures ANOVA. Finally, Tukey’s post hoc test was used. For comparing the groups with nonproven normality or with heterogeneous variances, the nonparametric Friedman ANOVA test and post hoc Friedman test were used. The post hoc power of the tests used was checked for the analysis of each parameter. A statistical test power less than 80% was considered an invalid outcome, and constructive conclusions were not formulated.

## Figures and Tables

**Figure 1 ijms-22-13604-f001:**
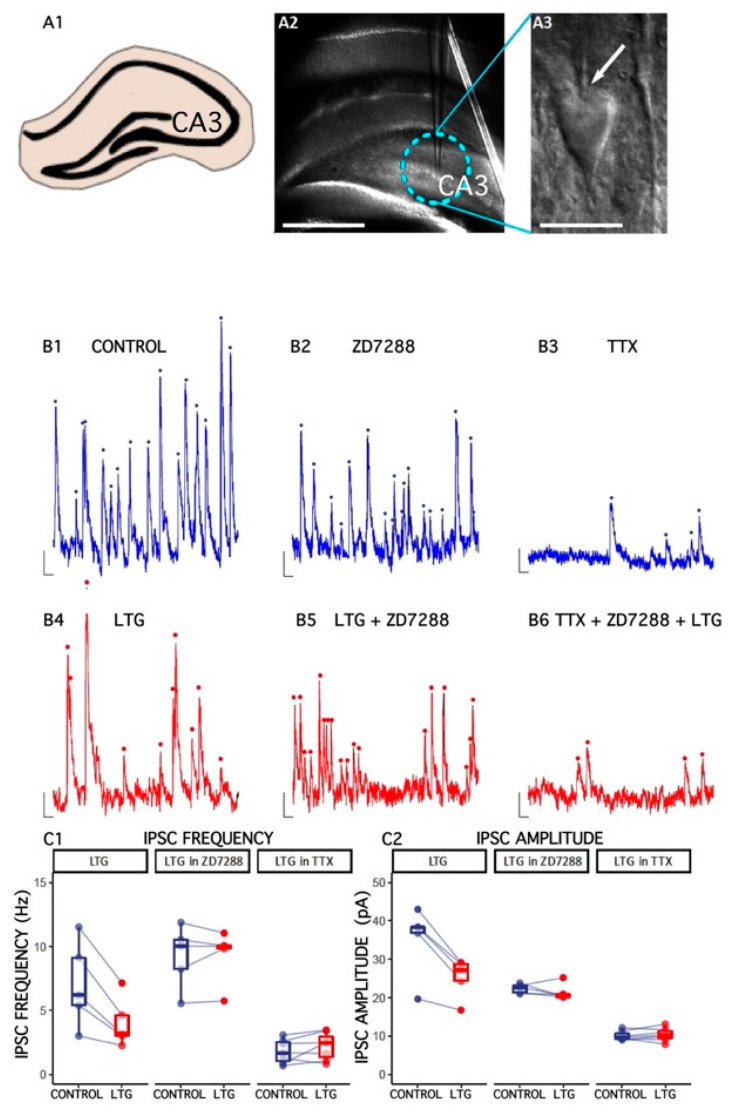
The effect of lamotrigine (LTG) on inhibitory synaptic transmission in CA3c. (**A1**) A schematic drawing of a rat HPC slice and CA3 area. (**A2**) A microscopic image illustrating the electrode placement in the HPC CA3c area. (**A3**) A photograph of a CA3c slice (DIC optics) showing a pyramidal cell with the electrode (an arrow). Scale bar: 500 μm. (**B1**–**B6**) Representative traces of sIPSC/mIPSC recordings from pyramidal cells before (CONTROL) and 15 min after LTG (50 μM) application to artificial cerebrospinal fluid (ACSF). Counted events are marked by dots. (**C1**,**C2**) Boxplots (boxes show first and third quantiles, the line shows the median; whiskers indicate the range corresponding to 1.5 times the interquartile range) showing the median sIPSC/mIPSC frequency and amplitude. Dots connected with lines correspond to individual neurons recorded before and after the application of LTG. Scale bars: A2 = 500 μm, A3 = 20 μm, B1–B6 = 10 pA, 100 ms; see the Results.

**Figure 2 ijms-22-13604-f002:**
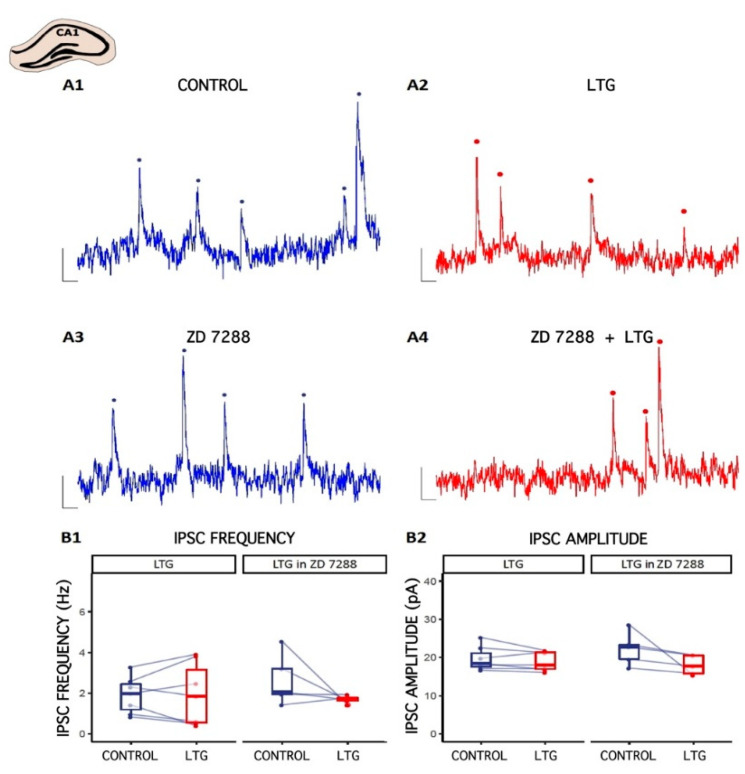
The effect of Lamotrigine (LTG) on inhibitory synaptic transmission in CA1. (**A1**–**A4**) Representative traces of sIPSC recordings from pyramidal cells before (CONTROL) and 15 min after LTG (50 μM) application to the artificial cerebrospinal fluid (ACSF). Counted events are marked by dots. (**B1**,**B2**) Boxplots (box shows first and third quantile, the line shows the median; whiskers indicate the range within 1.5 times the interquartile range) showing median sIPSC frequency and amplitude. Dots connected with lines correspond to individual neurons before and after the application of LTG. Scale bars: A1–A4 = 10 pA, 100 ms; see the Results.

**Figure 3 ijms-22-13604-f003:**
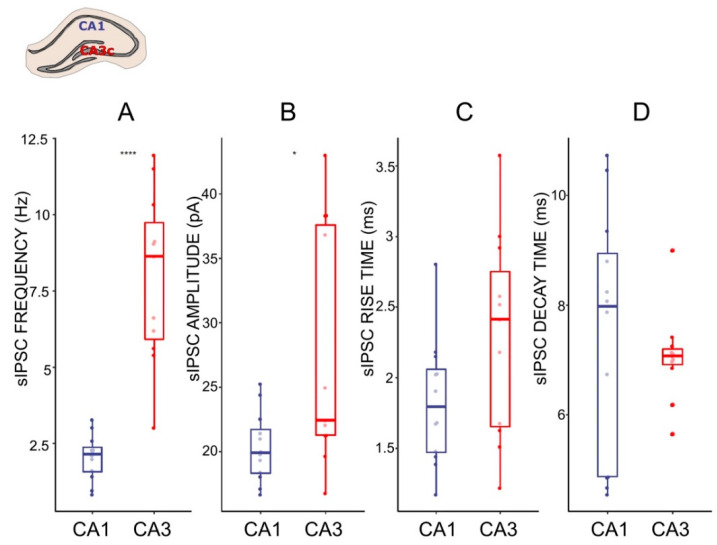
Characteristics of sIPSC in CA1 and CA3 pyramidal cells. (**A**–**D**) Boxplots (box shows first and third quantile, the line shows the median; whiskers indicate the range within 1.5 times the interquartile range) showing median sIPSC frequency, amplitude, rise time, and decay time, respectively. Dots correspond to individual neurons in normal ACSF. * *p* < 0.05, **** *p* < 0.0001; unpaired *t*-test and Wilcoxon Signed Rank Test respectively; see the Results.

**Figure 4 ijms-22-13604-f004:**
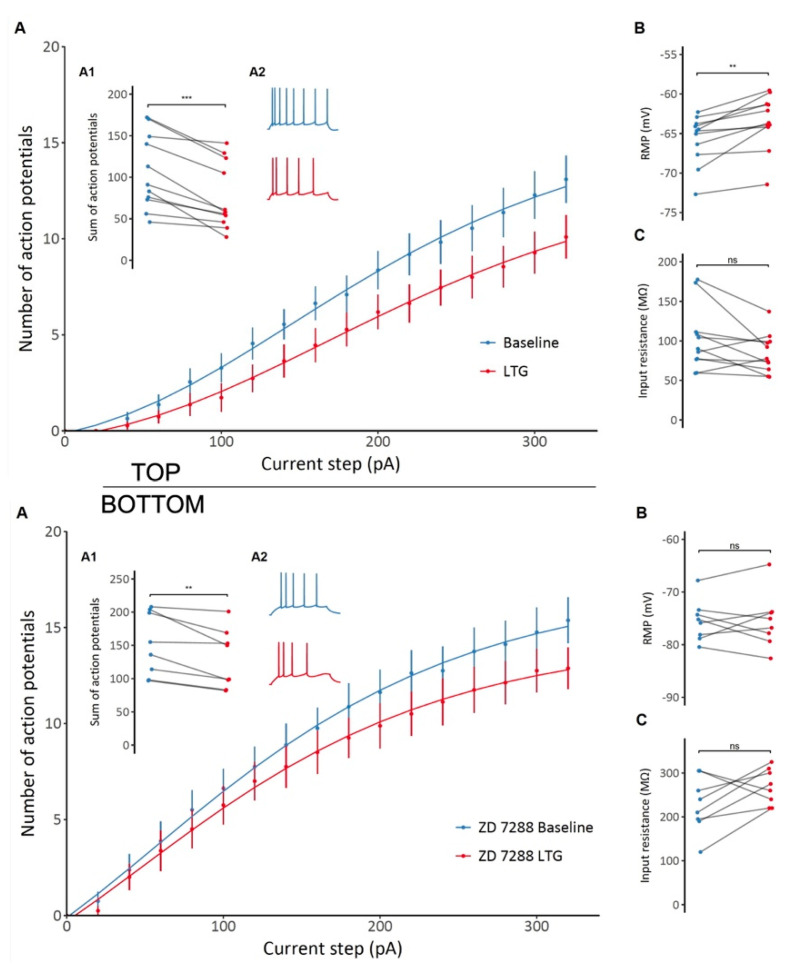
TOP: The effect of lamotrigine (LTG) on CA1 pyramidal neuron excitability and membrane properties. (**A**) A decrease in CA1 pyramidal neuron excitability is observed after LTG administration, as shown by the different F/I relationships before and after administration. (**A1**) A significant decrease in the total number of action potentials generated by a particular neuron in response to a series of depolarizing current pulses after LTG treatment (**A2**) Representative voltage responses to depolarizing current steps before and after LTG administration. (**B**) An overall depolarizing effect of LTG on CA1 pyramidal cells. (**C**) No significant changes in input resistance were observed; see the Results. BOTTOM: The effect of lamotrigine (LTG) on CA1 pyramidal neuron excitability and membrane properties in the presence of HCN selective blocker ZD7288. (**A**) A decrease in CA1 pyramidal neuron excitability induced by LTG in the presence of ZD 7288, as shown by the different F/I relationships before and after administration. (**A1**) A significant decrease in the total number of action potentials generated by a particular neuron in response to a series of depolarizing current pulses after LTG treatment in the presence of ZD 7288. (**A2**) Representative voltage responses to depolarizing current steps before and after LTG administration. (**B**) ZD 7288-induced prevention of LTG-induced depolarization. (**C**) No significant (ns) changes in input resistance were observed. *** *p* < 0.001, ** *p* < 0.05, paired *t*-test; see the Results.

**Figure 5 ijms-22-13604-f005:**
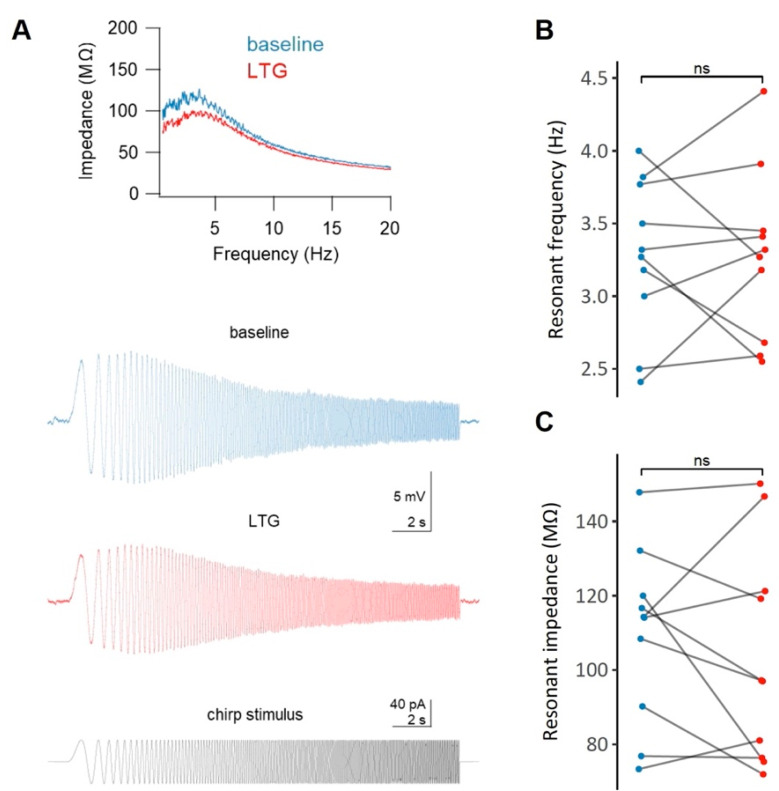
The effect of lamotrigine (LTG) on the membrane resonance properties of CA1 pyramidal neurons. (**A**) Representative impedance profiles (top panel) for a CA1 pyramidal cell before (blue) and after (red) the administration of LTG, along with the raw voltage traces (middle panel) and the chirp stimulus waveform (bottom panel). No significant (ns) changes in either the resonant frequency (**B**) or in the resonant impedance (**C**) after LTG treatment were observe; see the Results.

**Figure 6 ijms-22-13604-f006:**
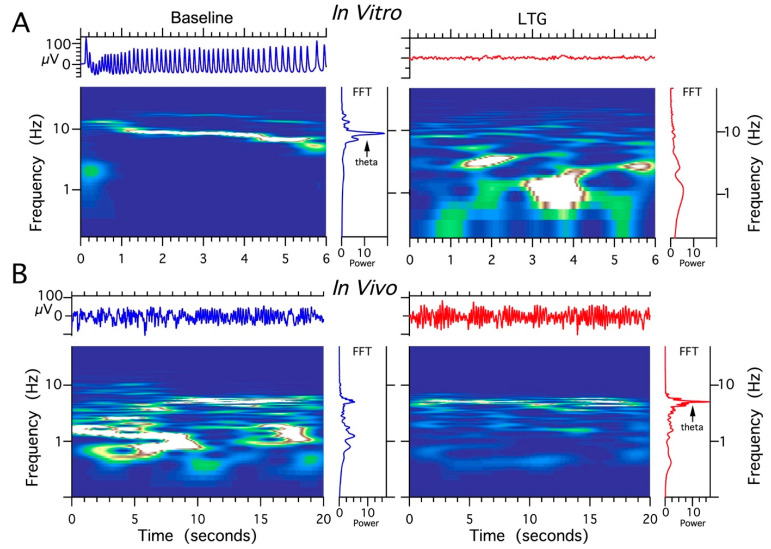
The comparison of lamotrigine (LTG) effect on local field potentials (LFPs) recorded in vivo vs. in vitro. (**A**) Abolishing effect of LTG treatment (100 μM) on HPC theta rhythms induced by 50 μM carbachol (CCH) in acute HPC slices. (**B**) The facilitating effect of LTG on spontaneous HPC theta rhythms in anesthetized rat (4 μg/1 μL).

**Figure 7 ijms-22-13604-f007:**
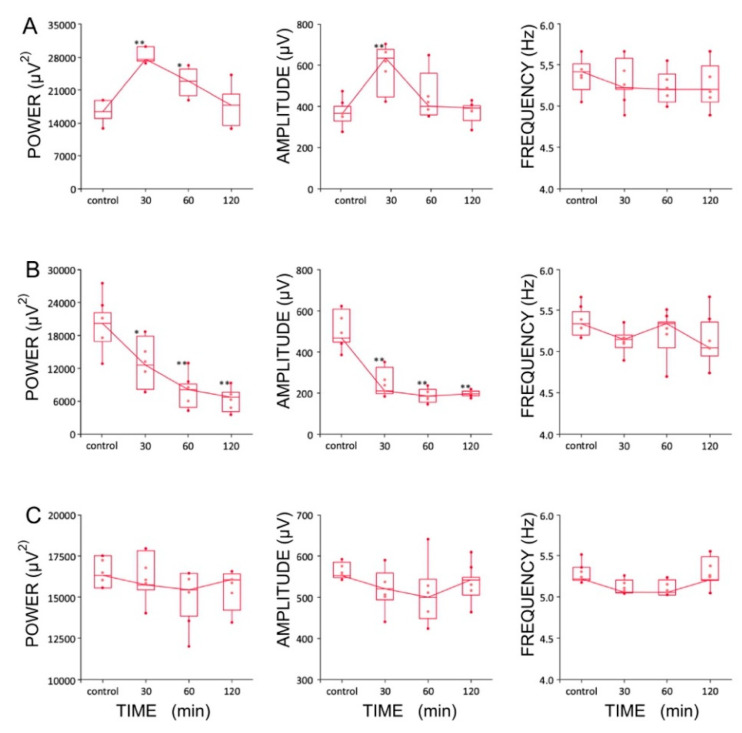
Boxplots showing changes in the parameters of the hippocampal theta rhythms after the injection of (**A**) lamotrigine (LTG, 4 μg/1 μL), (**B**) ZD7288 (4 μg/1 μL), and (**C**) a combination of LTG and ZD7288 (significance of differences: * *p* < 0.01 and ** *p* < 0.001). Data are presented as medians (horizontal lines) with lower and upper interquartile ranges. Whiskers indicate the range corresponding to 1.5 times the interquartile range.

**Table 1 ijms-22-13604-t001:** Summated quantitative details concerning power, amplitude, and frequency of the hippocampal theta rhythm after local injection of different agents (group I–IV).

		Parameters of Theta Rhythm	Groups			
			Group I LTG 4 µg/1 µL	Group II LTG 6 µg/1 µL	Group III ZD7288 4 µg/1 µL	Group IV ZD7288 4 µg/1 µL + LTG 4 µg/ 1 µL
preinjection	(control)	power (μV^2^)	16,396.2 (14,969.0, 18,789.1)	17,233.2 (15,846.6, 23,448.4)	20,023.1 (16,914.2, 22,104.3)	16,334.5 (15,567.1, 17,532.8)
		amplitude (μV)	565.5 (527.2, 598.3)	554.6 (487.2, 602.4)	468.8 (449.6, 609.1)	552.2 (547.6, 585,1)
		frequency (Hz)	5.4 (5.2, 5.5)	5.2 (5.1, 5.2)	5.3 (5.2, 5.5)	5.2 (5.2, 5.4)
postinjection	30 min	power (μV^2^)	27,502.8 (27,215.2, 30,145.6) *p* < 0.001	5125.5 (4264.8, 5383.1) *p* < 0.001	12,604.7 (8150.6, 17,800.0) *p* < 0.01	15,752.3 (15,444.6, 17,843.5)
		amplitude (μV)	833.2 (646.0, 874.6) *p* < 0.001	299.3 (287.2, 310.6) *p* < 0.001	210.4 (197.6, 325.5) *p* < 0.001	520.4 (493.3, 559.1)
		frequency (Hz)	5.2 (5.2, 5.6)	5.1 (5.1, 5.2)	5.1 (5.0, 5.2)	5,1 (5.1, 5.2)
	60 min	power (μV^2^)	22,903.4 (19,791.6, 25,458.7) *p* < 0.01	4854.2 (4684.6, 4978.5) *p* < 0.001	8160.7 (4880.0, 9121.9) *p* < 0.001	15,461.6 (13,852.4, 16,455.2)
		amplitude (μV)	598.6 (557.4, 760.1)	275.4 (258.2, 280.7) *p* < 0.001	186.6 (155.4, 218.6) *p* < 0.001	500.2 (448.1, 554.0)
		frequency (Hz)	5.2 (5.1, 5.4)	5.2 (5.1, 5.2)	5.3 (5.0, 5.4)	5,1 (5.0, 5.2)
	120 min	power (μV^2^)	17,755.4 (13,413.3, 20,123.1)	4807.6 (3988.4, 4993.8) *p* < 0.001	6802.0 (4134.1, 7634.5) *p* < 0.001	16,075.3 (14,224.8, 16,413.0)
		amplitude (μV)	591.2 (530.0, 602.5)	253.8 (242.6, 306.7) *p* < 0.001	197.3 (187.1, 209.2) *p* < 0.001	542.3 (505.4, 549.6)
		frequency (Hz)	5.2 (5.1, 5.5)	5.2 (5.1, 5.4)	5.0 (4.8, 5.4)	5,2 (5.2, 5.5)

Data were expressed as median and lower-upper quartile range (25–75%), *n* = 5 animals/group. Significance of differences estimated by ANOVA with repeated measurements tests and post-hoc Tukey tests or non-parametric Friedman ANOVA tests and post-hoc for Friedman.

## Data Availability

Not applicable.

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
