# Peer review of "Lamotrigine Attenuates Neuronal Excitability, Depresses GABA Synaptic Inhibition, and Modulates Theta Rhythms in Rat Hippocampus"

_ijms, 2021, doi:10.3390/ijms222413604_

Round 1

Reviewer 1 Report

The manuscript describes experiments assessing the effects of the antiseizure medication lamotrigine on neuronal electrical activity in in-vitro hippocampal acute slices and in-vivo obtained recordings. As such, this work does not appear to fit well the scope of the journal, which is mainly based on molecular studies (“...an advanced forum for molecular studies in biology and chemistry, with a strong emphasis on molecular biology and molecular medicine”). Results could be of interest, but they do not appear to be particularly robust since based on small numbers (around 5). The statistical analysis is mostly unclear: authors used a parametric test (Student’s t) for non-normally distributed data (as suggested by boxplots in figures). Other comments:

  • The use of abbreviations is inconsistent (HPC).
  • In introduction authors should mention the relevance of theta activity for epilepsy (see Costa et al. 2020 in Cell. Physiol. Biochem.).
  • Anesthesia may have confounded results obtained in vivo.
  • Discussion could be improved by considering the previous, related literature (for instance, Inaba et al. 2006, D’Antuono et al. 2007, both in Neuropharmacology).
  • Could the lack of connections of CA3 with the entorhinal cortex have influenced the different findings in CA3 and CA1 (as reported in D’Antuono et al. 2002, J. Neurophysiol., and Panuccio et al. 2010, Neurobiol. Dis.)? Indeed, inputs from the entorhinal cortex are preserved only in ventral slices.

Author Response

Thank you for the helpful comments to improve our manuscript.  All points have been addressed and revisions made (red type) in our revised manuscript.

Reviewer 2 Report

The study of P. Kazmierska-Grebowska and coauthors demonstrate the effects of a well-known antiepileptic drug, lamotrigine, on the excitatory and inhibitory neurotransmission in the rat hippocampal slices. Results are unambiguous and well described. The manuscript can be recommended for publication after minor revision:

  1. Some figures are disproportional and have to be corrected (especially the size of panels and fonts). I would recommend combining some figures (or even sections/paragraphs), for example, Figures 4 and 5.
  2. Discussion section is too long; it is better to make it more compact.
  3. The term “activator” may be inappropriate for the description of lamotrigine action on HCN channels since only several works demonstrate this effect. The information about the action on HCN channels has not been included in databases, yet (PubChem, IUPHAR etc.) 

Author Response

Reviewer 2:

Thank you for the helpful comments to improve our manuscript.  All points have been addressed and revisions made (red type) in our revised manuscript.

The study of P. Kazmierska-Grebowska and coauthors demonstrate the effects of a well-known antiepileptic drug, lamotrigine, on the excitatory and inhibitory neurotransmission in the rat hippocampal slices. Results are unambiguous and well described. The manuscript can be recommended for publication after minor revision:

            Thank you for these kind comments and suggested improvements:

  1. Some figures are disproportional and have to be corrected (especially the size of panels and fonts). I would recommend combining some figures (or even sections/paragraphs), for example, Figures 4 and 5.

Panels and fonts have been corrected for Figs 3 and 8 (new 7).  Figures 4 and 5 have been combined.

  1. Discussion section is too long; it is better to make it more compact.

We were not able to shorten our discussion because Reviewer 1 asked for more information to be added.

  1. The term “activator” may be inappropriate for the description of lamotrigine action on HCN channels since only several works demonstrate this effect. The information about the action on HCN channels has not been included in databases, yet (PubChem, IUPHAR etc.) 

Thanks for this suggestion, we have changed the term ‘activator’ to ‘modulator’ throughout the manuscript.

Round 2

Reviewer 1 Report

I thank the authors for the answers to my comments, I have no further concerns.